

# Exploring the transition between water and wind-dominated landscapes in Deep Springs, California as an analog for transitioning landscapes on Mars

Taylor Dorn, Mackenzie Day

Department of Earth, Planetary and Space Sciences, University of California Los Angeles, Los Angeles, 90095, USA

*Correspondence to*: Taylor Dorn (planetarytaylor@g.ucla.edu)

**Abstract.** Many planetary surfaces have been shaped by aeolian and fluvial processes, and understanding the resulting landscape is of critical importance to understanding changes in climate. Surface features on Earth and Mars are commonly observed using a variety of remote sensing methods. The observed geomorphology provides evidence of present- and paleo-processes, but interpretations are limited by the resolution of the data and similarity to well-understood systems on Earth. In this work, we study a complex fluvio-lacustrine and aeolian landscape at Deep Springs playa, California, using field measurements and remote sensing as an analog for a wet-to-dry transitioning landscape on Mars. The playa system in arid Deep Springs reflects fluvio-lacustrine processes in its interior, but transitions to aeolian-dominated processes along the playa margin. Weather station data and field observations collected over 34 months illustrate the interplay between aeolian and lacustrine processes and provide context for interpreting the observed geomorphology in aerial images. Our results showed a consistent distal-to-proximal geomorphic transition in the landscape defined by the changing expression of polygonal fractures, wave ripples, and evaporite deposits. Crescent shaped sedimentary deposits, originally suspected to be related to barchan dunes, proved unrelated to aeolian processes. We discuss the processes, sedimentary features, and climate drivers at Deep Springs to provide a potential framework for identifying and interpreting similar interactions between fluvio-lacustrine and aeolian geomorphology elsewhere on Earth, on Mars, and beyond.

## 1 Introduction

Planetary surfaces are shaped by aeolian, fluvial, glacial, volcanic, and other processes that drive erosion and deposition and define landscape evolution. When a single type of surface process dominates, the resulting landscape is well known. Glaciers carve U-shaped valleys, wind creates dune fields, water causes meandering rivers, and so on. The diversity of landscapes on Earth reflects the diversity of surface





processes, both individually and in combination, and is strongly influenced by changes in climate (Perron,
2017). Often when the climate shifts, the dominant landscape-shaping process shifts as well and
signatures of this change can be incorporated into the rock record (e.g., Tiedemann et al., 1989; Battarbee,
2000; Scherer et al., 2007;).

The morphologies and strata characteristic of a landscape transitioning from one dominant process to
another have been studied (e.g., Bull, 1991; Slaymaker et al., 2009; Fryirs and Brierley, 2012; Essefi et
al., 2014), but are less well established than those formed by one process alone. However, such
interactions are of critical importance to understanding climate change and process. On Mars, climate
conditions were once similar to Earth (Carr, 1982; Wordsworth, 2016), with a thick atmosphere, flowing
water, and complex fluvio-lacustrine landscapes (Irwin et al., 2005; Hynek et al., 2010). Several billion
years ago, Mars shifted away from Earth-like climate and sedimentary conditions to become the arid,
aeolian-dominated planet we see today (Greeley et al., 1993; Bridges et al., 2007; Grotzinger and
Milliken, 2010; Martínez et al., 2017; McLennan et al., 2018).  Previous research into this global transition
has studied the geologic record on Mars using images collected from the surface (e.g., Squyers et al.,
2006; Grotzinger et al., 2015) and from orbit (e.g., Milliken et al., 2010; Warner et al., 2010). For example,
strata observed in Gale crater by the Mars Science Laboratory Rover, Curiosity, record basal lacustrine
mudstones overlain by aeolian dune sandstones (Banham et al., 2018; Edgar et al., 2020; Rapin et al.,
2021). In Barth crater, deposits from an ancient crater lake transition laterally into aeolian dune strata
along the paleolake margins (Davis et al., 2019; Day et al., 2019). In this work, we study a complex wet-
to-dry landscape using field measurements and remote sensing as an analog for a transitioning landscape
on Mars. The playa system in arid Deep Springs, California, reflects fluvio-lacustrine processes in its
interior, but transitions to aeolian-dominated processes along the playa margin. We monitored Deep
Springs over 10 field visits spanning 34 months to study what surface changes occur during the wet and
dry seasons, the interaction between aeolian and lacustrine processes, and explore how playas on Earth
may be analogs for paleo-environments on Mars.





## 1.1 Deep Springs Playa

Deep Springs Playa (37.28° N, -118.04° W) sits between the White and Inyo mountains at an elevation of ~1524 m, ~24 km east of Big Pine, California (Fig. 1a). The ~5 km$^2$ Deep Springs Playa lies at the southeastern end of Deep Springs Valley which slopes 0.66° from the north to the south. Deep Springs

Playa consists of Deep Springs Lake, the perennially inundated southeastern region of the playa that varies in lateral extent throughout the year, and an outer surrounding drier playa region to the north and west with varying surface features and evaporite deposits (Jones, 1965; Lustig, 1965;). The mineralogy and hydrogeochemistry of Deep Springs are marked by a diverse range of saline and carbonate minerals that have received significantly more attention than the region's geomorphology (Peterson et al., 1963;

Meister et al., 2011; Plon, 2014). Coverage of evaporites on the surface of Deep Springs Playa changes as precipitation and groundwater fluctuate.

The water that keeps Deep Springs Lake perennially wetted is derived from runoff from the surrounding mountains and the fault zone springs to the east (e.g., Buckhorn and Corral Springs; Jones, 1965). Most

of the water flowing from Wyman- and Crooked Creeks to the north of the playa is collected for human use (Jones, 1965). Corral and Buckhorn Springs located to the east the playa have a combined total flow rate of 2.84 cubic feet/second (Jones, 1965). The flow rate provided for Corral and Buckhorn Springs, given the time of year, the limited data, and the locations they were measured in should be considered the minimum mean values for spring flow into the lake, though it is unclear how much the flow rate has

changed since the 1970s.

The White and Inyo mountains bounding Deep Springs Playa are broken up into normal fault blocks with fractures oriented in various directions (Miller, 1928; Lee et al., 2001). The valley between these mountains is filled by deep alluvial deposits almost certainly covering additional faults running through

the region (Miller, 1928). The most prominent fault is the Deep Springs Fault that runs along the eastern side of Deep Springs Valley for at least 24 km; marked by a fault scarp that rises sharply from the valley floor. The scarp is marked by sharp, triangular facets in its bottom 300 ft., above which, the scarp becomes more rounded indicating erosion and a pause in displacement (Miller, 1928).





## 2. Methodology

To study the interface between water-dominated and wind-dominated surface processes, we monitored weather and surface geomorphological features across Deep Springs playa. Weather data, surface imaging, drone imaging, and sediment were collected over 10 field visits between August 2019 and May 2022 (Fig. 1).

Weather data, including wind speed, wind direction, air temperature, soil temperature, precipitation, relative humidity, and barometric pressure, were collected using a NovaLynx 110 WS-25 modular weather station located in the western region of the playa (Fig. 1b). Wind direction and wind speed sensors were placed 2.5 meters above the surface, the air temperature sensor was placed 2 meters above the surface, and the soil temperature probe was placed 15 centimeters below the surface. Between May 1st, 2021 and May 11th, 2022, each instrument on the weather station continuously collected data at 1-minute sampling intervals, except for wind speed which instead took an averaged wind speed over the same 1-minute intervals. Data collected by the soil temperature probe included occasional spikes in recorded values that represent erroneous data. To ignore the influence of these erroneous records, we only report soil temperature measurements that are within three standard deviations of the mean daily soil temperature.

Three Apeman wildlife trail cameras were placed on the playa's surface and aimed towards the center of the playa to examine visual changes to the surface, including identifying any evidence of sediment transport and changes in the playa's surface texture over time. Trail camera images were collected every 5 minutes from March 2020 to November 2020 with a 720p image resolution. Two types of sediments traps and a push core were used during the study to estimate the sediment flux and grain size distribution in Deep Springs. Four stationary sediment traps constructed from PVC, each with three openings to capture sediment at heights of 8, 15, and 23 centimeters, were deployed from March 2020 to August 2020 and oriented to capture sediment coming from the east (Fig. 1c). When the stationary sediment traps captured no measurable sediment, they were replaced with rotating sediment traps that could align with





the wind. These modified Wilson and Cooke sediment traps (MWAC; Wilson and Cooke,1980) enabled aeolian sediment to be captured from any direction and had inlets to capture sediment at heights of 5, 15, 24, and 31 centimeters. The MWAC traps were deployed from October 2021 to May 2022 with a spacing

of 50 m, starting at the edge of the playa and ending at the weather station (Fig. 1d).

Drone images provided high resolution aerial views of the study area and were collected during each visit to the field area. A DJI Mavic Mini was used to image Deep Springs playa, including over the perennially wet Deep Springs Lake where the water was clear enough to see to the Lake floor. The drone was flown

at heights of 10-80 m and has a video resolution of 1080 p at 60 fps. Drone footage was taken 7 times between October 2020 and May 2022, providing coverage during each season of the year. Satellite images from the National Agriculture Imagery Program (NAIP) with a spatial resolution of 1 m/px and GoogleEarth were used to observe surface changes over time. These images were used to determine where and when surface features appear on Deep Springs, map features of interest, and see if a change in the

Deep Springs Lake shoreline position could be detected. Playa features in Deep Springs were mapped using ArcMap 10.6.

To understand how the Lake level at Deep Springs responds to local weather, we estimated the evaporation rate (E) using measurements from the weather station and the relationship (Chow et al.,

130  1988):

$$E = B(e_s - e_d).$$

$$\tag{1}$$

Where E is the evaporation rate (m/s), B is the vapor transfer coefficient (m Pa$^{-1}$ s$^{-1}$), $e_s$ is the saturation vapor pressure at ambient temperature (Pa), and $e_d$ is the actual vapor pressure (Pa). $e_s$ is determined using the relationship (Chow et al., 1988):

$$e_s = 611e^{\left(\frac{17.27*T}{237.3+T}\right)}$$





**(2)**

Where T is the air temperature (Celsius). The actual vapor pressure $e_d$ is given by:

$$e_d = e_s(Rh)$$

**(3)**

Where $Rh$ is relative humidity (%; Chow et al., 1988). Finally, B is given as:

$$B = \frac{0.622k^2 \rho_a u}{P \rho_w \left[ \ln \left( \frac{Z}{Z_0} \right) \right]^2}.$$

**(4)**

Where k is the Von Karman constant (0.4), $\rho_a$ is the density of air (assumed to be 1.579 kg/m$^3$), $u$ is the wind velocity (m/s) measured at height Z (250 cm), P is the atmospheric pressure (Pa), $\rho_w$ is the water density (assumed to be 1027 kg/m$^3$ for the briny water in Deep Springs Lake), and $Z_0$ is the roughness

height (assumed to be 0.03 cm) (Chow et al., 1988). Thus, the evaporation rate E depends on the quantities T, Rh, u, and P, each of which was recorded by the weather station in Deep Springs. Shoreline position is determined by a combination of inputs (e.g., precipitation, spring input, runoff) and outputs (e.g., evaporation). We estimate the rate of lateral shoreline movement ($\Delta L$) due to local climate controls (precipitation and evaporation) as:


$$\Delta L = \frac{P_r - E_{day}}{\tan(\theta)}$$

**(5)**

Where $P_r$ is the precipitation rate (mm/day) and $E_{day}$ is the total evaporation occurring in a given day

(determined by multiplying the rate from Equation 1 by the sampling interval of 1 minute, and then



summing for each 24 hour day). We use a surface slope across the playa, $\Theta$, of 0.21°, calculated using a 10 m spatial resolution USGS digital elevation model in ArcMap 10.6.

## 3. Results

Deep Springs playa was studied between August 2019 and May 2022 with surface and drone imaging, sediment trap data, and field observations collected during 10 field visits. Weather data were continuously collected between May 1, 2021 and May 11, 2022.

### 3.1 In-situ weather observations

Air temperature, soil temperature, precipitation, barometric pressure, relative humidity, and wind direction were recorded at 1-minute intervals between May 1, 2021 and May 11, 2022 (Fig. 2). Reported wind speed measurements represent an average wind speed over the same 1-minute intervals between the same dates.

The air temperature at Deep Springs playa averaged 11.4° C with a standard deviation of -4.9° C (hereafter, we report data as the mean ± 1 standard deviation) between May 1, 2021 and May 11, 2022; reaching a high temperature of 41° C on July 11, 2021 and a low temperature of --16.3° C on February 24, 2022 (Fig. 2b). The mean daily temperature difference, calculated by averaging the difference in the daily minimum and maximum temperatures, was 5.8° C. The greatest difference in the daily minimum

temperature and daily maximum temperature was 18.1° C on March 1, 2022. The smallest difference in temperature was -11.3° F on December 23, 2021. Soil temperatures in Deep Springs playa followed the same daily diurnal and seasonal trends as observed in the air temperature measurements (Fig. 2b), however, these trends lagged behind air temperature by ~2 weeks. The soil temperature overall averaged 14.4° C ± -2.3° C, where the highest soil temperature was 40.8° C on July 25, 2021 and the lowest soil

temperature was -6.6° CCCCC on December 18, 2021.

Deep Springs playa received a total of 994 mm of precipitation from May 1, 2021 and May 11, 2022 (Fig. 2a). The highest 24-hour period of precipitation occurred on April 19, 2022 where the playa received a





total of 108 mm of rain. For the majority of the observed days, no precipitation occurred on the playa; out

of a total of 376 days of data collection, 257 days received no precipitation. The playa received 67% (668

mm) of its total of 994 mm of precipitation between the months of April and June. The remaining days

of the year received an average rainfall of 1.2 mm/day ± 3.5 mm/day.

Barometric pressure averaged 1,051.1 mbar ± 166.7 mbar and ranged between 835.2 mbar and 1,211.5

mbar for the monitoring period (Fig. 2b). Humidity ranged from 1.3% to 99.8% with an average humidity

of 34.9% ± 24.2% (Fig. 2a). Humidity increased during the winter months, November through January,

where the average humidity was 65.1% ± 11.9%. In the remaining days of the year, the mean daily

humidity reached >60% a total of three days; on July 30, 2021, October 25, 2021, and March 29, 2022.

Wind in Deep Springs was northerly, blowing from a mean wind direction of 7.7°± 98.7°, at an average

speed of 2.6 m/s ± 2.5 m/s (Fig. 2c, d). For 41.7% of the observed time period, wind emanated from

between 360° and 45° (i.e., northerly to northeasterly) (Fig. 2c, d). Northwesterly, more variable, and

overall higher wind speeds occurred through the summer months, where, between June 1 and September

30, the mean daily wind speed was 2.8 m/s and had a mean direction of 351.3° ± 25.2°. During the winter

months, October through February, the winds were northerly, less variable, and had an overall lower wind

speed with a mean wind direction of 13.8° ± 18.1° and a daily mean wind speed of 1.9 m/s. The maximum

1-minute-averaged wind speed was recorded on October 25, 2021 as 23.2 m/s. Separately, the highest

recorded wind gust of 26.4 m/s was recorded on April 30, 2022 (Fig. 2c). There was no significant

correlation between wind speed and wind direction (linear regression $R^2$ <0.01) (Fig. 2c).


Based on equations 1-4 above, Deep Springs playa totalled 2.8 m of evaporation during the weather

monitoring period. The mean evaporation per day was 7.38 mm/day ± 13.8 mm/day (Fig. 2a). The

maximum calculated amount of evaporation in a day was 176 mm on June 13, 2021. Evaporation is

greatest in the summer months when the temperatures and wind speeds are highest. Conversely,

evaporation is at its lowest from late Fall into Spring as the temperature and wind speed decrease (Fig.

2a).



## 3.2 Surface features observed from satellite images

Satellite images with sufficient resolution to resolve morphologies on the playa surface were available to
view using GoogleEarth from July 1993, August 1998, December 2005, June 2012, May 2013, August
2013, and May 2020 and NAIP imagery from July 2010, June 2012, August 2014, June 2016, September
2018, and July 2020. In all satellite imagery it was not possible to confidently identify the shoreline
position. Flow features were observed outside of Deep Springs Lake indicating that water has been
continually active along the transition between Deep Springs Lake and Deep Springs playa. Ripples in
the northwest region of the playa are first observed in May 2013 and have persisted in this region since
this time. Polygonal fractures were observed in every image in some region of the playa. Their occurrence
was primarily within the basin interior and not limited to a particular time of year. The size of the polygons
varied in each image ranging from ~10 m to ~30 m wide.

Crescentic features, half-moon shaped forms with distinct, light-toned banding going across each feature,
were observed in a 2013 satellite image near the edge of the maximum observed shoreline level (Fig. 3c-
e). This was the only satellite image in which these features were present. A total of 217 crescent features
were measured and mapped resulting in an average horn to horn width of 5.6 m ± 2.4 m. These forms
were observed within Deep Springs Lake abutted against the northern and western margins of the Lake.
Each feature was several meters in scale and identifiable in satellite images by the bright surface albedo
with respect to the surrounding playa. Each feature contained concentric, nested lineations that followed
the outline of the crescent shape. The orientation of the crescents followed the shoreline, with crescentic
forms on the western side of Deep Springs Lake opening towards the west, and crescentic forms in the
northwest opening towards the northwest. All other crescentic features between these two regions had
their open ends similarly oriented towards the direction of the shoreline. Some of the crescent features
were isolated, whereas others were touching, connected, and not discrete forms from one another (Fig.
3c-e). It is unclear, even in this high resolution image if the features were under water or not.



## 3.3 Field observations

During each field visit to the study area, the surface and surroundings of the playa were observed. Multiple surface features were observed in both Deep Springs Lake and Deep Springs playa, in between these two regions is the fluctuating shoreline which is marked by a series of crenulated edges (Fig. 3c, 5c, 7d, f). The shoreline is bounded by a small berm that creates a topographic barrier, but the water level in the basin interior fluctuates throughout the year. Crescent features resembling those measured in the 2013 satellite image were observed in January 2021 (Fig. 3a-b). Whereas the crescent features observed in satellite imagery were bright toned, these features match the surrounding muddy surface in coloration and are identifiable by some white evaporite deposits and periodic linear features. These smaller crescentic features were observed in the same location as those studied in satellite images: near the western edge of the maximum observed shoreline level of Deep Springs Lake with their open ends oriented towards the west.

Linear features with wavelengths ranging from ~5 cm to ~2 m were observed on the surface in several locations primarily in north- and southwest regions of the playa (Fig. 4a, b). The larger linear features (Fig. 4d, e), measuring ~1-2 m in wavelength, were observed in drone footage during each drone survey and have their crests oriented towards the northeast, perpendicular to the dominant wind direction in Deep Springs. Smaller linear features, viewed on foot in the southwest region of the playa, ranged from 5 cm to ~1 m in wavelength (Fig. 4a, b). During the colder and wetter months, linear features found in this region of the playa had shorter wavelengths ranging from 5 to 30 cm (Fig. 4a), whereas during the warmer and drier months, linear features found in the same area displayed longer wavelengths (Fig. 4b) ranging from ~50 cm to ~1 m. Along with an increase in wavelength, when the temperature increased and the air became drier during the summer months, the linear features observed on foot were observed to be greater in height and cracked along their crests where their interiors were found to be hollow (Fig. 4c).

Springs, formed when subsurface pressure forces groundwater to the surface, are identifiable in Deep Springs Lake by the orange-to-purple deposits surrounding the spring outlet (Fig. 5a, b). It is unclear when these springs become active or inactive as the dark deposits around the springs can be seen at all





times of year, but do not persist in the same location. Spring deposits range in scale from a few centimeters across, to more than a meter, with the larger features further into the basin interior and only visible in satellite and drone imaging (Fig. 5a). The spring deposits interrupt and superpose the polygonal fractures on the surface of the playa, which vary in size from 2 cm – 7 m in diameter. The polygons are primarily hexagonal and are not bound to any one region of the playa. The differences in polygon morphology parallel the differences in surface texture moving distal to proximal into the basin. At the outer edge of the playa, hardpacked, smooth mud is occasionally disrupted by sharp-edged polygonal fractures (Fig. 7a). Further into the basin the surface transitions to a rougher surface texture with centimeter-scale relief and more muted polygons across the still muddy surface (Fig. 7b). Continuing into the basin, the surface roughens further, creating a popcorn texture associated with the powdery evaporite deposits, efflorescent salts precipitated as the water evaporates (Fig. 7c). The texture varies laterally in this zone of the playa, with some locations supporting a surface crust of mixed mud and evaporite cements (Fig. 7d). The efflorescent evaporite texture was observed only during the cool, wet months and was ~5 to 8 cm in thickness, while the rest of the year was dominated by the thin (~1 cm or less), hard mud-evaporite crust (Fig. 7d). The water content of the playa mud increases toward the center of the basin, making walking across the playa increasingly difficult. At the shoreline of the playa Lake, the point where standing water is present on the surface, the surface is marked by crystalline evaporites that form a several millimetre-thick crust on the surface that is easily scraped away revealing the brown silt and mud underneath. These crystalline deposits exhibit lineations that parallel the shoreline and no mud is incorporated into the crystalline material. Beyond this shoreline, polygonal fractures are still present, but the edges of the polygons are defined by upwardly tilted slabs of the evaporite crust. In some places water ponds in the interior of the polygons or submerges them entirely. Near the shoreline, some polygons are exposed subaerially, composed of the surface crust above the standing water, and show coherent crystals in their interiors, precipitate from further evaporation (Fig. 7e). In several of the runoff channels in the southern and southwestern region of the playa, as well as other local depressions individual evaporite crystals were observed (Fig. 7e). These did not coat the surface like the thin white crust or the powdery, flour-like surface. Instead, these crystals were found sporadically with larger crystals found in deeper depressions.



Sediment in the study area was assessed visually, with sediment traps, and using a 2.54 cm PVC push core. In each set of sediment traps, over two different deployment timespans, no sand was collected at any height. In the first sediment traps that were deployed (Fig. 1c) only small amounts of dust which were too insignificant to measure, was collected. In the second set of sediment traps (Fig. 1d) no sediment of any size was captured. Two push cores were collected to a depth of 75 cm and 60 cm on the western and eastern margins of the playa, respectively (Fig. 1a). The cores compressed to 30 cm and 24 cm, respectively, and in both cases were composed of massive, black, silt and clay-sized mud (Fig. 6h). By manual inspection, no sand-sized grains were found in either core, consistent with observations of the surface and from the sediment traps. Loose sediment besides the powdery evaporite deposits was rarely observed on the mud-flat surface of the playa. In August, 2019, well-rounded clasts ranging from 500 um to >2 mm in diameter were seen in groups on the playa surface, concentrating in local topographic lows. On inspection these grains could be crushed between two fingers and were determined to be aggregates of silt and clay grains (Fig. 6g). Beyond the playa margins the landscape transitions from mud flats to grasses and shrubbery. Here, vegetated mounds previously interpreted as coppice dunes (Jones, 1965) stand 1-2 m high. Some sand was observed in the coppice dune interior, distal from the playa itself.

### 3.4 Evaporation in Deep Springs

The shoreline of Deep Springs Lake fluctuates throughout the year as precipitation, surface runoff, and evaporation rates change. The shoreline is at its furthest extent during the wet season, when precipitation and surface runoff outpace evaporation, and at its lowest just after the dry season, when evaporation exceeds precipitation and groundwater. As the playa transitions from wet to dry, the shoreline slowly recedes towards the east. Between May 2021 and May 2022, we calculated a total evaporation of 2.77 m of water depth, which translated to net lateral shoreline retreat of 494.2 m based on climate measurements alone.

### 4. Discussion

The facies progression in the Deep Springs study area follows the common pattern associate with playa systems on Earth: bounding topography edged with alluvial fans yields to sand flats, then to mudflats,




then saline mudflats, and finally a saline pan (Rosen, 1994; Warren, 2016). The transition from hardpack
mud to crystalline encrusted saline pond is made clear by the changing geomorphology at the surface,
which in turn reflects the interaction between water and wind across the playa. For example, the polygonal
fractures that criss-cross the surface in every satellite image require both wet and dry processes to form.
Polygonal fractures are commonly found in muddy sediments of tidal flats, lacustrine shorelines, playa
lakes, and fluvial floodplains (Plummer and Gostin, 1981). The fractures form subaerially when water
evaporates from the pore space of the mud, creating volume loss and tensile stress that causes the substrate
to fracture (Neal et al., 1968, Weinberger, 1999). In Deep Springs, polygonal fractures reflect the
interaction between water saturation and evaporation across the basin, and their changing morphology,
from sharp-edged in the mudflat margins, to rough and platy in the pan interior, shows the increased role
of evaporite precipitation into the basin interior. Similarly, the changes in surface texture distal to
proximal across the basin are mostly identified by the changing amount and style of evaporite deposits.
The cracked, dry-mud surface gives way to an undular surface with efflorescent, powdery evaporites, that
transitions into increasingly dominant evaporite deposits until reaching the Lake shoreline where salts
have crystalized across the surface. We attribute these changes to the increase in water saturation and
associated increase in evaporite deposition as the water levels fluctuate and pore water evaporates.


One interesting difference between Deep Springs and other playa-lake systems on Earth is the absence of
sand and sandy bedforms in the playa interior. No sand was observed or captured in sediment traps during
the observation period, despite the maximum wind speed of 23.2 m/s (51.9 mph), which far exceeds the
threshold of motion for quartzo-feldspathic sand (Dong et al., 2002). Coppice dunes west of the playa
margins suggest sand does reach the region but is trapped by the vegetation at the playa margin (Jones,
1965). Vegetation increases surface roughness, causing sand to be deposited when wind speeds decrease
as the wind passes across the convoluted surface (Wolfe and Nickling, 1993; Lancaster and Baas, 1998).
In Deep Springs, the playa is surrounded by grasses and shrubs that anchor coppice dunes to the west and
yield to reedy marshes around the springs to the north and east. Based on the lack of measurable sand in
the playa interior, we interpret that the local vegetation is preventing sand from reaching the basin.





The linear ridge patterns observed in the surface further reflect the interaction between water and wind. Straight-crested, with centimeter- to meter-scale wavelengths, these features share the morphology of ripples and are presumably formed either by wind or water (Clifton and Dingler, 1984; Nishimori and

Ouchi, 1993). Observed many times during the study, when the ripples were measured, their orientation was perpendicular to the dominant wind direction as recorded by the weather station. This would suggest that the features were aeolian ripples, however the features are composed entirely of silt- and clay-sized grains, with no sand present. Furthermore, the surface mud was cohesive, and although the bedforms were always observed subaerially exposed, no movement in the bedforms was ever observed in the field. If not

formed by particles moving in wind, the ripples must be formed subaqueously, either as current or wave ripples. Small, low relief channels at the margins of the playa suggest that some overland flow into the basin does occur and would be a possible driver of current ripples. However, the ripples observed were not confined to the topographic lows of the channels and were not oriented to suggest flow to the basin interior. The coincidence of ripple orientation and the perpendicular dominant wind direction, suggests

wind played some role in the ripple formation, making wind-driven waves the most likely driver of these features. Symmetry across the crests of the ripples could further suggest a wave-driven origin (Evans, 1941; Clifton and Dingler, 1984), but the rough texture of the ripple surface, in part related to evaporite precipitation, makes the crestline symmetry difficult to assess with confidence. We interpret that the abundant ripples formed as wave ripples when the playa was occasionally flooded. As the water receded,

the ripples remained, too cohesive to be resurfaced by aeolian processes, and eventually were loosely cemented in place by evaporite minerals. The smaller ripples in the southwestern region of the playa did not persist over the duration of the study, suggesting that the cementing minerals redissolved and the bedforms were erased during episodes of subsequent rain or flooding.

**4.1 Shoreline change and water budget**

The progression in the landscape from hardpacked mud at the margins of the playa, to crystalline evaporite crust in the interior of the playa was consistent during every field visit, but the extents of each texture varied between visits, raising the question: is the system as a whole drying, filling, or in dynamic steady state? Using the measured inputs from precipitation and losses from evaporation, we estimated that during





our monitoring period, Deep Springs Lake lost approximately 2.77 m of equivalent water depth,
       translating to 494.2 m of shoreline retreat. These measurements would suggest that Deep Springs Lake is
       net drying and might completely dry over the next several years. However, our measurements do not
       account for the inputs from springs (e.g., Corral Spring and Buckhorn Spring), or from overland flow.
       Accounts in previous research (Miller, 1928, Jones, 1965) suggest that the overall extent of Deep Springs
playa has not changed over the past roughly 100 years. Allowing for seasonal fluctuations, this means
       that the external inputs to Deep Springs Lake, from springs, streams, and overland flow, must balance the
       net loss calculated in this work, and suggests that the system as a whole is in dynamic steady state. This
       interpretation is supported by observations of satellite images that show the shoreline changing, but show
       no overall change in the extent of the playa over time.


## 4.2 Barchan-shaped crescent features, but not aeolian dunes

       Many of the surface features observed in Deep Springs playa are well-understood and common to
       evaporitic playa environments. However, the crescent-shaped surface features are more enigmatic and
       subject to debate. The crescent shaped sedimentary deposits were first noticed in a 2013 satellite image
where their high albedo and barchan-dune-like morphology suggested wind may be transporting the
       abundant evaporite minerals (Fig. 3c). The sizes and shapes of the Deep Springs crescent features
       qualitatively and quantitatively match the sizes and morphologies of barchan dunes in plan-view (Hersen,
       2004; Parteli et al., 2007). The concentric, nested lineations seen in the Deep Springs features are not
       consistent with active bedforms but are consistent with the plan-view expression of cross strata left behind
by migrating bedforms (Hunter, 1977; Rubin, 1987). If barchan dunes had developed on the playa, the
       observed bright crescent patterns could be explained by cementation of the bottommost layer of dune
       sand and subsequent removal of the overlying material by stronger winds. This would expose only the
       basal strata in plan-view across the playa, as observed in Figure 3e. The interpretation that the crescent
       features originated from barchan dunes is supported by the interactions observed between crescents,
similar to interacting dunes (Kocurek et al., 2010). However, several observations put the interpretation
       of these features as barchan dunes into question. First, the crescent orientations are not consistent with
       the dominant northerly winds. Dunes forming in the measured wind regime should develop crescents that





open to the south and southwest, but the bright crescent features in the 2013 satellite image open toward the west and northwest, perpendicular to the local shoreline of the playa Lake. Furthermore, aeolian dunes

develop from saltating sands and at no point during the study was saltation observed on the playa and no sand was captured in sediment traps. That some transient conditions led to the deposition of sand on the playa is possible, but in order to develop crescent features of that scale from barchan dunes, the associated dunes would have needed to be, in some places, more than a meter tall (Parteli et al., 2007).

Similar features to the crescents identified in the 2013 Deep Springs satellite image have been studied on the Makgadikgadi and Ntwetwe pans in Botswana (Burrough et al., 2012; McFarlane and Long, 2015; Franchi et al., 2020). Here, crescent shaped features with concentric, nested lineations form a field across the playa with individual features linking and interacting, as well as forming at a variety of sizes. The features in Botswana are notably larger than those in Deep Springs (several hundred meters in scale rather

than several meters), but as in Deep Springs, the orientation of the features is consistently in the same direction and their morphology is similar to barchan dunes. Previous work has debated the origin of the features in Botswana, with some interpreting the features as basal strata from aeolian paleo-barchan dunes (Burrough et al., 2012), and others attributing the curvilinear features to spring mound formation in the salt pan (McFarlane and Long, 2015; Franchi et al., 2020).


Based on evidence collected in our study, we suggest that the crescents in Deep Springs cannot have formed from aeolian dunes. The morphological similarity to barchan dunes is striking, however, the lack of sand on the playa, the dissimilar orientation with respect to the wind regime, and the coincidence of the orientation with the local shoreline, suggest that water on the playa, rather than wind, is driving the

formation of these features. Given that an aeolian origin is unlikely, we consider whether the spring mound interpretation of features in Botswana could apply to the features in Deep Springs. Several named springs are perennial around the study area, but smaller ephemeral springs are also present in the basin interior (Fig. 3b, 5). Pressurized groundwater from springs can sometimes carry with it sediment that is then deposited in the immediate vicinity of the spring, creating slight topography (Roberts and Mitchell,

1987; Franchi et al., 2020). In Deep Springs, an alternative explanation for the barchan-shaped features





invokes slight topography from the spring mounds, receding water levels, and evaporite precipitation. In this case, the nested curvilinear patterns that define the crescent shape would be strandlines, analogous to bathtub rings, with high-albedo mineral precipitation defining the position of the water level on the topographic mound as the water level falls (Fig. 8). Asymmetry in the topography of the spring mound,

relatively steeper on the upslope side and shallower toward the basin interior, would create the barchan shape. Rather than marking the migration of a dune from east to west, each curve would instead mark the retreat of the surface water level from west to east.

Variations in topography across the playa surface are below the resolution limit of available digital

elevation models. However, based on field observations, the playa surface is not perfectly flat, and surface undulations of <10 cm are present, defining local channels and mounds that could help define strandlines as the water levels change. The largest springs in the system were always beyond the shoreline and inaccessible on foot but were captured in drone surveys. Figure 5a shows examples of the springs in the Lake interior, clearly visible by the dark rust-colored deposits around each. Figures 3b and 9d shows

examples of asymmetrical ring deposits submerged beneath the water surface, but superposing the polygonally fractured salts. Other examples of asymmetry around the springs (Fig. 5b) suggest inhomogeneous deposition around the mouth of the spring that could lead to low-relief topography.

In January 2021, a drone survey during a field visit captured patterns of ripples on the surface, the outline

of which formed crescent shapes at the surface (Fig. 3a-b). Unlike the bright features observed in satellite imagery, with the high albedo curves in the feature interior that followed the orientation of the crescent, these features are identifiable only by the ripple terminations and ripples remaining straight-crested and parallel. The orientation of the ripples is, as before, approximately perpendicular to the dominant wind direction, suggesting wind did play a role in shaping the ripples. Why their extent terminates to form a

barchan-shape is unclear but could be attributed to small changes in topography causing small differences in water depth and therefore, wave ripple formation. The crescent-bounded ripple patches seen in this survey are morphologically distinct from the bright crescentic features in the 2013 satellite image and may be an example of coincident barchan-shaped features formed via different processes.



### 4.3 Deep Springs playa as an analog for Mars

Evaporite deposits on Mars have been identified from orbital spectrometry and from surface measurements (see reviews in Ehlmann and Edwards, 2014; Rampe et al., 2020). Similarly, paleo-lake and ancient playa systems have been interpreted from Mars geomorphology (Grotzinger et al., 2015; Davis et al., 2016). In particular, intra-crater lakes were once abundant on Mars (Goudge et al, 2012) and with no liquid water present on the surface of Mars today, each one of these lake systems must have dried and become a salt pan at some point. The processes occurring in Deep Springs playa must have also occurred on Mars at various points in the ancient past, making the morphologies observed in this study potentially applicable to interpretations of features on Mars. Previous work has suggested many terrestrial analogs for facies assemblages on modern and ancient Mars (e.g., Marlow et al., 2008; West et al., 2010; Xiao et al., 2017). We suggest that Deep Springs may be a suitable terrestrial analog for drying playa regions on the paleo-martian surface, including, as an example, in Meridiani Planum (Squyres et al., 2006). In both Deep Springs and Meridiani Planum, the exploration area of the Mars Exploration Rover Opportunity, evaporite deposits define a surface with abundant polygonal fracture patterns (Fig. 9c). Additionally, the curvilinear features tracing shorelines that can be seen in Deep Springs (Fig. 9b) can also be found in Meridiani Planum (Fig. 9a). The scale of the surface features are similar in Deep Springs and on Mars, despite an age difference of at least 1 billion years. Although no one feature is sufficient to claim that Meridiani Planum and Deep Springs are analogous environments, the many features seen on both suggest that the surface of Deep Springs as seen in the field may provide insights into the paleo-character of evaporitic surfaces, like Meridiani Planum, at the hand-sample and field observation scale.

The crescentic features observed in satellite image of Deep Springs highlight a potential source of uncertainty in interpreting geomorphology of Mars. Sharing the morphology of barchan dunes, including apparent cross strata exposed in plan-view, the crescent features observed in Deep Springs represent an example of aeolian-dune-like surface morphologies that could incorrectly be interpreted from satellite images as aeolian in origin without sufficient context. In Deep Springs, the wind measurements collected in the field and the coincidence of the crescent orientations with the local shoreline both provide strong



evidence that the dune-like features are not aeolian in origin, but such contextual information may not be available in examples of this morphology on Mars. Crescent-shaped features at a similar scale to barchan dunes are resolvable in high-resolution images of Mars (Fig. 9a), and although sometimes there is

sufficient geologic context and lateral extent to determine that the features were formed by dunes (Anderson et al., 2018; Day et al., 2019), exposures can be resurfaced, eroded, or covered, obscuring the geologic context needed to disambiguate these features. The observations of this work suggests additional caution is advisable when interpreting curvilinear, barchan-shaped features in plan view that are associated with drying playa environments.


One critical difference between Earth and Mars is the role of life. Microbial communities are presumably active in Deep Springs playa, as is common in playa environments elsewhere on Earth (e.g., Navarro et al., 2009; Makhdoumi-Kakhki et al., 2012; Sanz-Montero, 2013), and tracks from macrofauna, including insects, mammals, and birds, were seen across the playa surface. No plant life was present in the playa

interior, but at the margins, particularly near the major springs, vegetation obscured the surface textures that are easily studied in the basin interior. Coppice dunes to the west are anchored by vegetation and necessarily not present on Mars. The absence of sand-sized grains in the playa interior suggests that vegetation at the playa margins may work to hinder sand transport, preventing larger grains from reaching the playa interior. With no vegetation on Mars, one might expect sand to play a larger role in playa

sedimentation.

Other than the vegetated coppice dunes, none of the morphologies observed in Deep Springs strictly require the presence of life. Playa systems like Deep Springs are known on Earth to harbour diverse populations of microbial life (Navarro et al., 2009; Makhdoumi-Kakhki et al., 2012) and could be

potential areas of astrobiological interest for potential halophilic life in ancient environments on Mars (Kunte, 2002; Oren et al., 2014). The specific mineralogy present in the playa was not tested in this work, but has been the subject of many previous investigations (e.g., Peterson et al., 1963; Meister et al., 2011; Plon, 2014), which determined Deep Springs playa to be rich in carbonate, sulfate, and halide minerals (Fig. 9f). On Mars, similar mineral assemblages have been observed and interpreted to reflect similar





depositional settings (Fig. 9e; see reviews in Niles et al., 2013; Ehlmann and Edwards, 2014). Whether the assemblage of minerals in Deep Springs and other playa systems, including on Mars, could have been sufficient to support microbial life on its own is outside the scope of this work. Instead, we propose the system in Deep Springs as a geomorphological analog for similar paleo-environments that must have existed on Mars and suggest that they may be of potential interest to future astrobiologists.


## 5. Conclusions

The surface of Deep Springs playa is shaped by tightly coupled wetting- and drying processes, leading to a suite of surface features observed widely on Earth and Mars. Polygonal fractures and evaporite deposits reflect the interaction between water saturation and evaporation across the basin. The cracked, dry-mud

surface at the distal edge of the basin gives way to an undular surface with efflorescent, powdery evaporites toward the basin interior, that transitions into increasingly dominant evaporite deposits until reaching the Lake shoreline where salts have crystalized across the surface. These basin-wide changes are attributed to a basin-ward increase in water saturation and evaporite deposition as the water levels fluctuate and pore water evaporates. Deep Springs is notable for an absence of sand and sandy bedforms

in the playa interior. Dunes to the west of the playa are anchored by vegetation and are downwind from the dominantly northeasterly winds. Linear patterns across the playa, with centimeter to meter-scale wavelengths, were interpreted as wave ripples. In a satellite image of Deep Springs, patterns of high-albedo banding were observed, matching the morphometry of barchan dune strata preserved in plan-view. However, the lack of saltating sand in the playa and the orientations of the features rule out an origin in

meter-scale aeolian dunes, as had been debated for similar features elsewhere on Earth. Instead, we interpreted the high-albedo, nested crescent-shapes to reflect strandlines around low relief spring mound topography. Formed from evaporite precipitation as the water level in Deep Springs Lake drops, the curvilinear banding resembles barchan dune cross strata in plan-view, but is unrelated. Similar curvilinear banding has been observed on Mars, and the results of this work suggest that the morphology alone is

insufficient to interpret these barchan-shaped features as paleo-dunes without additional context. The geomorphology and sedimentary processes observed in Deep Springs reflects a transition from wet to dry sedimentation across the basin. Similar basins on Mars once hosted lakes that dried in a major climate



shift. The patterns observed in Deep Springs reflect an environment that must have also been present in many locations on ancient Mars. The processes, sedimentary features, and associated weather conditions

described at Deep Springs provide a framework for identifying other interactions between fluvio-lacustrine and aeolian systems in images and data acquired from orbit and on the surface on Earth, Mars, and beyond.

## Acknowledgments

The authors thank Arturo Sotomayor, Joseph Witchalls, Jacob Widmer, Suzanna Gevorgyan, Jonathan Sneed, Martha Mejia, and Genevieve Dorn for their assistance in the field. [placeholder]





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

**Figures**



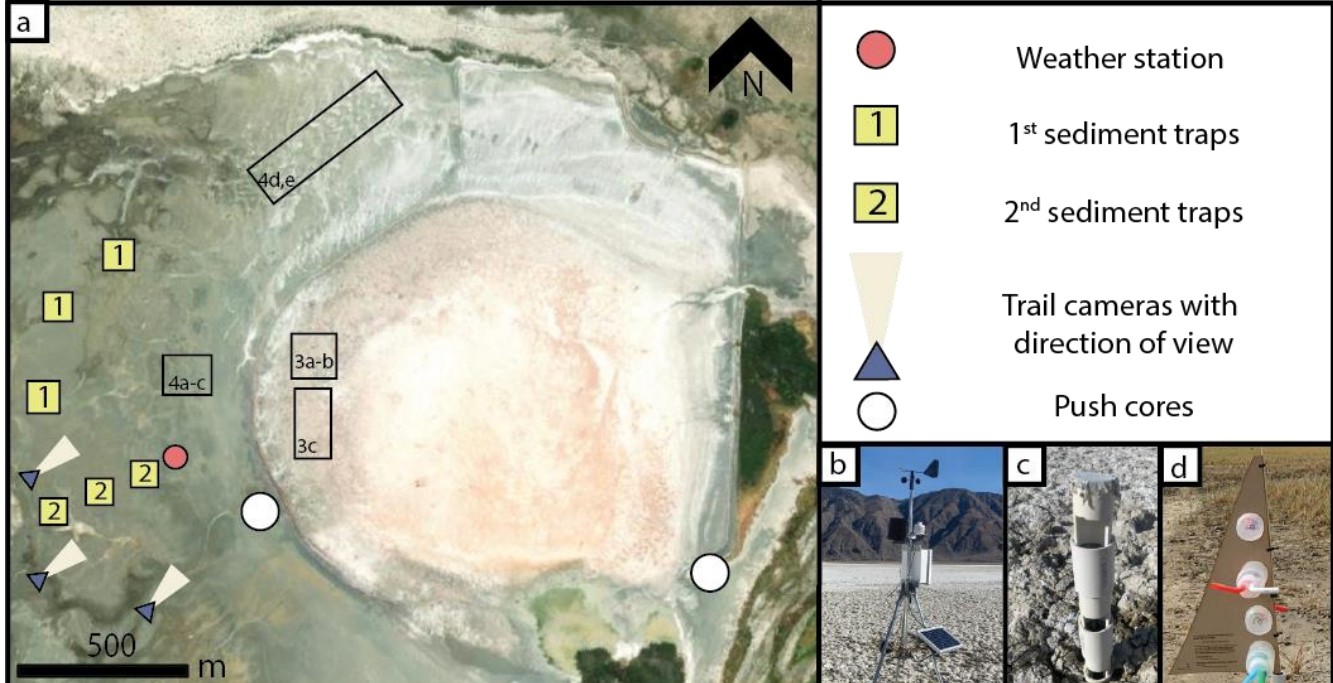

**Figure 1. Study area and monitoring sites, Deep Springs, CA**. (a) Extent of Deep Springs playa. Locations of other figures are boxed. Image courtesy of Google Earth (© Google Earth 2022). (b) Modular weather station indicated by red dot in (a) with data collected continuously between May 1, 2021 – May 11, 2022. (c-d) Sediment traps deployed from March 2020 to August 2020, indicated by the "1" in (a), and October 2021 to May 2022, indicated by the "2" in (a).








**Figure 2. Weather in Deep Springs, CA between May 1, 2021 and May 11, 2022** (a-b) 1-minute interval weather data. Precipitation is greatest during the spring and is dormant during the summer through winter. Soil and air temperatures follow the same general trend with soil temperatures lagging behind air temperatures by ~2 weeks. Humidity is at its greatest during the late fall and winter. (c) Time series of all wind directions and wind speeds experienced in Deep Springs playa. (d) Wind rose diagram showing predominantly northerly to

easterly winds in Deep Springs. Wind rose shows all wind speeds in Deep Springs ≧ 0.1 m/s. Along with a majority of the winds in Deep Springs emanating from the north to the east, higher wind speeds also primarily emanate from these same directions



**Figure 3. Crescent features observed in Deep Springs** (a-b) dark-toned crescent features captured on drone footage in January 2021. Features appear to be concentrated around the exposed mudcracks and are all oriented towards the west, aligning with the orientations observed in a satellite image in 2013. (c) Satellite image from 2013 showing an abundance of light-toned crescent shaped features oriented towards the west. These features, like those captured by drone in (a-b) are concentrated along the western edge of Deep Springs Lake. Banded shoreline of Deep Springs Lake is indicated by the bounding dashed black lines. (d) Open ended crescent features, indicated by black arrows, oriented in same direction. (e) Linked crescents (indicated by black arrow) near the shoreline of Deep Springs Lake. Images c–d courtesy of Google Earth (© Google Earth 2022).





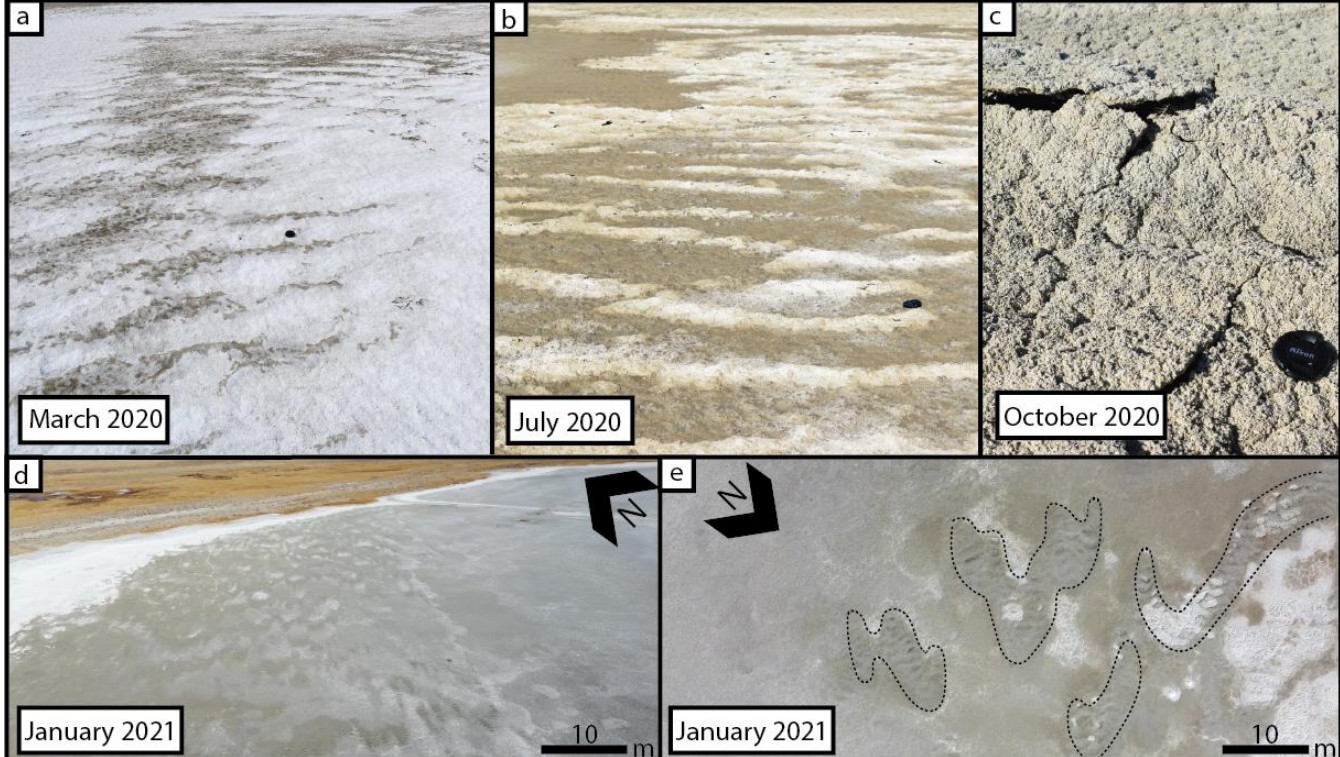

**Figure 4. Ripples observed around Deep Springs** (a-b) Ripples seen in Deep Springs during a wet (a) and dry (b) portion of the year. As the playa dries throughout the year, the ripples in the same region of the playa had longer wavelengths. (c) Ripples cracking during the warmest portion of the year. While in this condition, ripples become hollow in their interiors (d-e) Ripples observed via drone in the far north end of the playa. Ripples, outlined in dashed lines in (e) are within an overall crescent shape and are oriented in towards the southwest, downwind of the dominant wind direction in Deep Springs. Images d-e courtesy of Google Earth (© Google Earth 2022).






**Figure 5. Large-scale features observed by drone around Deep Springs** (a) Spring mounds surrounded by polygonal fractures within Deep Springs Lake. (b) Closer view of one spring mound and polygonal fracture. (c) Edge of Deep Springs Lake showing compressed shorelines that are similar to those seen in Figure 7d, f. The shoreline is indicated by bounding dashed lines. In between these dashed lines, the shoreline is filled with curvilinear features that are compressed and following the same pattern.





**Figure 6. Small-scale features observed around Deep Springs** (a) Mudcracks surrounding larger, raised evaporite encrusted polygonal fracture. (b) Curvilinear features often seen in drainage channels along edges of Deep Springs playa. (c) Smaller mudcracks often appearing on the surface of Deep Springs during the warmer months. (d) Edge of Deep Springs Lake showing curved shoreline (indicated by dashed black lines). (e) Evaporite crystals commonly found in several runoff channels to the south and southwest. (f) Deep Springs Lake shoreline showing a series of crenulated edges. (g) Loose, well-rounded clasts observed on the surface in August 2019, primarily in topographic lows. Clasts were easily crushed between fingers. (h) Two cores measuring 30 cm and 24 cm taken on the western and eastern margins of the playa. These cores did not have any visible sand-sized grains and were primarily composed of massive, black, silt and clay-sized mud.



**Figure 7. Surface texture changes across Deep Springs playa** (a-e) Show changes in surface texture moving from Deep Springs playa into Deep Springs Lake. (a) Large, flat, meter-scale polygonal fractures dominate the playa, especially in the drier outer regions that only become watered when it rains or snows (b) Rougher, slightly raised surface with no cracks (c) Raised polygonal fractures with curled edges with thin evaporites deposited primarily on these raised edges (d) Small, centimeter-scale mudcrack on the eastern side of Deep Springs Lake that often experiences wetting from nearby springs (e) Wettest accessible region of Deep Springs showing dark brown surface, polygonal fractures just beyond the edge of shoreline





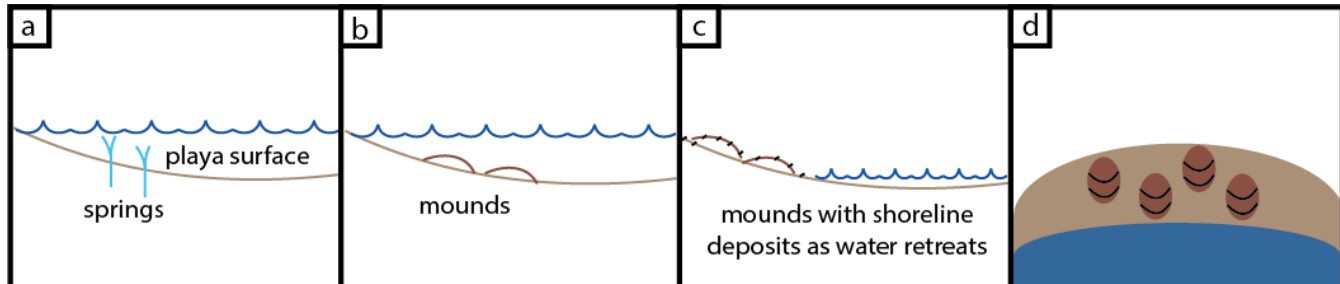

865

**Figure 8**. **Conceptual model for how crescent features are formed in Deep Springs.** a) The waterline extends past the springs located at the margin of the lake. b) Spring discharge carries mud and silt that gets deposited near the spring. c) As the waterline retreats, subaerially exposing the spring deposits evaporite precipitation marks strandlines on the mound topography. d) Plan view of the mounds after the shoreline has retreated leaving behind waterline marks on the mounds.

870

**Figure 9. Mars surface feature analogs to Deep Springs playa** (a) Terrain on Mars resembling the crescent features observed in Deep Springs (HiRISE image ESP_071800_1805), (b) Bright concentric features in Deep Springs resembling similar crescentic shaped features on Mars. Dashed black lines show nested, concentric lineations across each feature. Image courtesy of Google Earth (© Google Earth 2022). (c) Polygonal fractures found in the interior of Barth crater, a paleolake on Mars. (CTX image: G20_025939_1864_XI_06N334W) 875 (d) Polygonal fractures observed in Deep Springs as the playa's surface dries out (e) Mosaic of MAHLI images from sol 809 showing jarosite crystals (f) Evaporite crystals found in several depressions in Deep Springs.