# Peer review of "Exploring the transition between water and wind-dominated landscapes in Deep Springs, California as an analog for transitioning landscapes on Mars"

_EGUsphere, 2022_

## Author Comment (AC1)

**Intracrater sediment trapping and transport in Arabia Terra, Mars**

T.C. Dorn and M. D. Day

**Response to reviewers**

The editor highlights several areas where the manuscript could be improved in clarity and we have incorporated the suggested changes, each of which is described below. The original reviewer comments are shown in black, and our responses are in red below each comment.

**Comments to the author**:
Please use SI units in the manuscript throughout:
line 81: miles > km

Units were corrected from miles to kilometers

lines 181-191 and Fig 2 use Celsius not F

All units were changed to Celsius in text and in figure 2

line 311: inches to cm
Units were changed from inches to cm

Another two small items:
Line 388 resurfaces should be resurfaced

Resurfaces was stated in the corrected tense

line 876: should panel e be the same photo as shown in c?

Figure was corrected to not have duplicate image in panel "e" and "c"

---

## Author Response (AR1)

Response to reviewers – Deep Springs Manuscript

Reviewer #1 - Anonymous

The importance of this paper stems from the author's ability to link monthly and yearly weather data and changes in the landscape to long-lived wet and dry changes on Mars. Most terrestrial studies related to Mars were on a larger timescale, i.e., thousands or millions of years. The work is clearly novel, well-written, and perfectly appropriate for publication in ESURF after considering the following minor suggestions.
We appreciate the reviewer's support and have addressed each specific comment below.

A very interesting point in this work is that the authors found that Deep Springs lost 2.77 m of equivalent water depth, which indicates a 494.2 m shoreline retreat. They then concluded that the drying was net and that a complete drying could occur within the next few years.I wonder if the authors could extract a profile of the water changes over the past years to see whether the lake's drying occurs gradually or abruptly. I think this might be important for early Martian climate.
This is an interesting idea, but in the case of Deep Springs, the drying from evaporation that the reviewer notes is balanced by input from the local groundwater springs. We did look at satellite images from previous years, and note in the text that the water level did not change. The reviewer's suggestion is interesting, but cannot be done for this region because of the additional spring input.

Another point that could be interesting is that the Deep Springs occur over an arid zone, and the wet to arid geomorphic processes observed could be aligned with the recent studies that used paleolakes and valley network junction angles to infer that early Martian climate was likely arid to semi-arid (e.g., Stucky de Quay et al., 2020; Seybold et al., 2018). I would suggest calculating the aridity index of the region using rainfall and evapotranspiration. Then It would be great to build a discussion on whether the Martian playa lakes might have formed under arid to semi-arid climates.
As suggested, text was added to define and calculate an aridity index, report it for Deep Springs, and compare with the noted references for Mars.

I would suggest mentioning in the introduction that this study examines the transition from wet to arid at monthly and yearly timescales. This is important because most of the terrestrial studies linked to understanding the Martian surface are on a bigger timescale, i.e., thousands or millions of years.
The suggested change was made.

Somewhere in the discussion, it would be valuable to mention some morphometric characteristics based on Earth analogs that define similar playa lakes on Mars. Maybe in the section of Deep Springs playa as an analog for Mars.
The suggested change was made.

References

Stucky de Quay, G., Goudge, T. A., & Fassett, C. I. (2020). Precipitation and aridity constraints from paleolakes on early Mars. Geology, 48(12), 1189–1193. https://doi.org/10.1130/g47886.1

Seybold, H. J., Kite, E., & Kirchner, J. W. (2018). Branching geometry of valley networks on Mars and Earth and its implications for early Martian climate. Science Advances, 4(6). https://doi.org/10.1126/sciadv.aar6692
* * *
Reviewer #2 – Favaro

This paper provides us with a novel dataset of monthly and yearly changes to a transitioning landscape that can be used as an analogue for Mars. The authors have thoroughly investigated their study site and share a comprehensive dataset. I provide line-by-line minor comments, mostly having to do with clarification of language.
We appreciate the reviewer's support and have addressed each specific comment below.

Minor Comments:

Introduction

General: Somewhere in the introduction, I would highlight the temporal timescales of this study, as must Mars or Martian analogue studies consider much longer timescales. Having a monthly/yearly timestep is quite novel and should be highlighted.
The suggested change was made.

Line 27: Suggested change to "When a single type of surface process dominates, the resulting landscape is well known; for example,  glaciers carve U-shaped valleys, wind creates dune fields, water causes meandering rivers, and so on".
The suggested change was made.

Line 29: Delete "both individually and in combination with".
An important point of this work is that the results of different surface processes acting in combination are understudied with respect to the results of a single process alone. We leave this text to emphasize this point.

Line 36: The text states "However, such interactions are of critical importance to understanding climate change and process". Do you mean to understanding how climate change affects or modifies the dominant geomorphological processes affecting a landscape? This sentence needs to be flushed out a bit more.
As suggested, the sentences was revised for clarity.

Line 37: Suggested addition to "On Mars, climate and atmospheric conditions were once similar to Earth"…
Climate generally includes the atmosphere and we prefer this broader term.

Line 64: Suggest changing "significantly" to "substantively". You use 'significantly' later on in the paper (line 213) to refer to a statistically significant result.
The suggested change was made.

Line 72: Please convert flow rate to metric
The suggested change was made.

Line 72: Delete "provided". Could change to "The flow rate calculated by Jones (1965)…".
The suggested change was made.

Line 83: Please convert feet to meters.
The suggested change was made.

Methodology

Line 87: "Surface imaging": Satellite surface imaging or field photos? You've qualified that you have 'drone images', so you might want to qualify this as well.
As suggested, the text was revised to "field photographs".

Line 105: Suggestion: Add (Figure 1) at the end of the sentence.
The suggested change was made.

Line 107: 'sediment traps', not 'sediments traps'.
The suggested change was made.

Line 110: Suggestion: "However, when the stationary sediment traps…"
The suggested change was made.

Line 112: Suggest having "fig. 1d" come after your first mention of MWACs.
The suggested change was made.

Line 117: Suggest change from "collected during each visit o the field area" to "field site", just for variety.
The suggested change was made.

Line 120: Please spell out 'frames per second'.
The suggested change was made.

Line 120-121: Change "drone footage was taken 7 times" to "drone footage was collected…"
The suggested change was made.

Line 124: Change "…and see if a change" to "evaluate if".
The suggested change was made.

Results

Line 186: Please confirm the temperature is in Celsius and not Fahrenheit?
We confirm the temperatures are in Celsius.

Line 190: Only one C is necessary
Typographical error. The suggested change was made.

Line 216: For the first sentence, can you include the dates here? I know you just reported them, but there are a few dates floating around and this would help the reader know which 'monitoring period' is relevant.
The suggested change was made.

Line 227: Suggested change for the sentence starting "In all…": "Overall/Unfortunately, the resolution of the satellite images could not be used to confidently identify shoreline position."
This text was not changed as it is not a resolution issue, per se. The transparency of the water and similarity of submerged and exposed surface pattern makes the boundary unclear.

Line 231: Change "some region" to "various regions".
The suggested change was made.

Line 236: I was confused by your use of "maximum observed shoreline level". In lines 227-228, you state the satellite imagery couldn't confidently identify shoreline position. But here you suggest you can. In the lines above, do you mean you were unable to map the progression of shorelines? If that's the case, I would make that clearer.
Text was added to clarify that the maximum observed shoreline level is interpreted to be the position of the broadest extent of evaporite deposits as observed in satellite images and confirmed in the field. We interpret that surface precipitation of evaporites observable in satellite images only occurs with inundation, and therefore must reflect the extent of recent surface water.

Line 239: Lake does not need to be capitalized.
The suggested change was made.

Line 240: Change "in scale" to "across".
The suggested change was made.

Line 246-247: Suggested rewording: "Some of the crescent features were isolated, whereas others were touching or connected, and therefore not discrete forms (Fig. 3c-e)."
The suggested change was made.

Line 247: Add comma after "high resolution image".
The suggested change was made.

Line 251: Delete 'field'
The suggested change was made.

Line 252: Change "observed" to "documented" – you used 'observed' in the previous sentence.

The suggested change was made.

Line 252: Start a new sentence at "In between".
The suggested change was made.

Line 255: Suggested rewording "Crescent features resembling those identified and measured in 2013…"
The suggested change was made.

Line 256-257: Suggested rewording: "Whereas the crescent features observed in 2013 satellite imagery were bright toned, these the 2021 features match the surrounding muddy surface…". Adding the years helped me follow the narrative a bit better.
The suggested change was made.

Line 263: observed how? In situ/drone/satellite?
The text was changed to "imaged".

Line 267/Line 272: Change "on foot" to "in situ".
This change was not made so as to differentiate between observations made with a drone and from the surface, both of which are in situ.

Line 273: change "greater in height" to "taller".
The suggested change was made.

Line 277 onward: Suggested change: "It is unclear when these springs become active or inactive as the dark deposits around the springs can be seen throughout the year, but do not persist in the same location. Spring deposit diameters range in scale from a few centimeters to more than a meter across in, with larger features occurring further into the basin interior…"
The suggested change was made.

Line 290: change "in thickness" to "thick".
The suggested change was made.

Line 293: Which lake? Or just don't capitalize Lake.
The suggested change was made.

Line 294: This is super picky and very pedantic, but you've spelled meter differently here. Meter is the US spelling, while -metre is the French/Canadian/UK spelling. I only noticed because I mix mine up as well.
For consistency, all "meter" instances were changed to "metre" for this EGU publication.

Line 308: Change "…only small amounts of dust which were too insignificant to measure" to "trace amounts of dust".
The suggested change was made.

Line 313: what does "manual inspection" mean? In situobservations of the core materials?

We intend the most literal meaning here, by hand. The cores were broken up by hand and the mud in the cores was crushed between fingers to identify any sand sized grains (which skin can feel from grittiness). Visual observation alone might have meant we missed things, but by manually breaking up the core, we are sure there was no sand.

Line 326: Suggest adding "…groundwater inputs".
The suggested change was made.

Discussion

Line 332: Associated
The suggested change was made.

Line 353: Please convert mph to km/hr
Rather than report m/s, mph, and km/hr, we updated the text to simple report m/s.

Line 354: Threshold for motion for dry sand. I wonder how much more windspeed is needed for wet sand. Although, you have demonstrated there is no sand in the playa interior, but it's just something to think about
Agreed. Obviously, the thresholds of motion for wet sand are much higher, but what exactly they are is not well quantified and likely depends on how much moisture is present.

Line 353: Suggest adding "Straight-crested, and with centimeter…".
The suggested change was made (line 363 of original text).

Line 369: Delete "ever"
The suggested change was made.

Line 395: Suggest changing to "has not changed over roughly the past 100 years".
The suggested change was made.

Line 400: This was a very interesting section.
Thank you!

Line 480: Careful with definitive language like "must have dried and become a salt pan…". It did dry (obviously), but may not have become a salt pan. This is a nit-picky comment, I realize, but did want to share.
As suggested, the salt pan comment was removed.

Line 499: Change to "aeolian dune-like surface morphology"
The suggested change was made.

Line 507: "The observations from this work…".
The suggested change was made.